

# Coral affected by stony coral tissue loss disease can produce viable offspring

Sandra Mendoza Quiroz[1,2], Raúl Tecalco Renteria[1,2], Gandhi Germán Ramírez Tapia[1,2], Margaret W. Miller[1], Maria Victoria Grosso-Becerra[2] and Anastazia T. Banaszak[2]

[1] SECORE International, Miami, FL, United States of America
[2] Unidad Académica de Sistemas Arrecifales, Universidad Nacional Autónoma de México, Puerto Morelos, Quintana Roo, México

## ABSTRACT

Stony coral tissue loss disease (SCTLD) has caused high mortality of at least 25 coral species across the Caribbean, with *Pseudodiploria strigosa* being the second most affected species in the Mexican Caribbean. The resulting decreased abundance and colony density reduces the fertilization potential of SCTLD-susceptible species. Therefore, larval-based restoration could be of great benefit, though precautionary concerns about disease transmission may foster reluctance to implement this approach with SCTLD-susceptible species. We evaluated the performance of offspring obtained by crossing gametes of a healthy *P. strigosa* colony (100% apparently healthy tissue) with that of a colony affected by SCTLD (>50% tissue loss) and compared these with prior crosses between healthy parents. Fertilization and settlement were as high as prior crosses among healthy parents, and post-settlement survivorship over a year in outdoor tanks was 7.8%. After thirteen months, the diseased-parent recruits were outplanted to a degraded reef. Their survivorship was ~44% and their growth rate was 0.365 mm ± 1.29 SD per month. This study shows that even diseased parent colonies can be effective in assisted sexual reproduction for the restoration of species affected by SCTLD.

## INTRODUCTION

In the Caribbean, coral diseases have caused a drastic decline of more than 50–80% in live coral cover in recent decades (*Aronson & Precht, 2001*; *Weil, 2004*; *Jackson et al., 2014*). In 2014 a new coral threat emerged in Florida, the stony coral tissue loss disease (SCTLD) (*Precht et al., 2016*). Disease signs include multiple lesions with rapid tissue loss, often resulting in total colony mortality of highly susceptible species within a few weeks (*Alvarez-Filip et al., 2019*). Since its appearance, SCTLD has spread along Florida's Coral Reef to the Western Caribbean, the Greater Antilles, the Bahamian Archipelago, and the Eastern Caribbean (*AGRRA, 2019*). It has affected at least 25 species of scleractinian corals, many of which are reef-building species that contribute to architectural complexity (*Alvarez-Filip et al., 2022*). Total coral colony density has been reduced to less than 29% of baseline data in some cases (Mexican Caribbean) and severely affected species have been

Corresponding author
Anastazia T. Banaszak,
banaszak@cmarl.unam.mx

virtually extirpated from local sites (*e.g.*, *Precht et al., 2016*; *Alvarez-Filip et al., 2019*; *Neely et al., 2021*). Consequently, basic functions of the coral community, such as calcification, are greatly impaired (*Estrada-Saldívar et al., 2020*), threatening the physical persistence and ecological function of coral reefs at a regional scale (*Perry & Alvarez-Filip, 2019*).

Stressors, in general, are known to impair various phases of reproductive success. In corals, temperature stress and bleaching have been associated with depressed fecundity or spawning in subsequent years (*Fisch et al., 2019*), sometimes for multiple years (*Levitan et al., 2014*; *Johnston et al., 2020*), as well as reduced fertilization rates (*Omori et al., 2001*). Sedimentation and turbidity impair fertilization (*Ricardo et al., 2015*) and potentially a host of other reproductive processes (*Jones, Ricardo & Negri, 2015*). Nutrient pollution disrupts gametogenesis and fertilization (*Ward & Harrison, 2000*; *Harrison & Ward, 2001*; *Loya et al., 2004*). Physical breakage also precludes gametogenesis over subsequent years (*Lirman, 2000*). However, relatively little is published on the reproductive capacity of corals affected by disease. Two histology studies have shown reduced fecundity in marginal lesions of *Orbicella faveolata* colonies affected by chronic disease, but the unaffected tissues on the same colonies had similar fecundity as control colonies (*Weil, Cróquer & Urreiztieta, 2009* addressing Yellow-band disease; *Borger & Colley, 2010* addressing white plague). Disease-induced partial mortality in *Acropora palmata* did not affect fertilization rates or embryonic development of gametes collected, regardless of proximity to the lesions. However, egg volume was significantly lower in colonies with partial mortality in comparison to apparently healthy colonies (*Piñón González & Banaszak, 2018*).

The rapid declines in populations of key reef-building species and the reduction in colony density (*Estrada-Saldívar et al., 2021*) are expected to limit the success of sexual reproduction of these species. Additionally, low coral recruitment recorded after the SCTLD outbreak, with a predominance of brooders and opportunistic species, and the low cover of crustose coralline algae that facilitate coral recruitment (*Caballero-Aragón et al., 2020*), highlight the urgent need for interventions to restore coral populations. Coral 'rescue' efforts, focused on various species in unaffected areas have been initiated in several regions (*Grosso-Becerra et al., 2021*; *O'Neil et al., 2021*) to provide genetic resources for rebuilding these populations. However, the risks or the potential to leverage genetic resources from colonies that are already affected by SCTLD have not been previously addressed. Meanwhile, given that SCTLD is known to be highly transmissible (*Aeby et al., 2019*; *Meiling et al., 2021*) in laboratory settings, biosecurity concerns might discourage active restoration intervention. Additionally, there is no evidence as to whether SCTLD-affected colonies can produce viable gametes.

*Pseudodiploria strigosa* is the second most affected species by SCTLD in the Mexican Caribbean (*Alvarez-Filip et al., 2019*), causing grave concerns regarding its capacity for recovery and focusing attention on intervention strategies to encourage species recovery (*e.g.*, *Grosso-Becerra et al., 2021*). It is a hermaphroditic broadcast spawner that is expected to spawn six to eight nights after the full moon in August and/or September in the Mexican Caribbean (*Jordan, 2018*) with horizontal transmission of photosynthetic symbionts. In this study we determined the viability of offspring obtained from an apparently healthy colony (0% tissue loss) crossed with a colony affected by stony coral tissue loss disease

(more than 50% tissue loss) of *P. strigosa* and compared the results with those of offspring from unaffected parents obtained in previous years.

## MATERIALS & METHODS

Spawning surveillance of 13 colonies of *Pseudodiploria strigosa* in La Bocana Reef (375 m$^{-2}$ monitored area) in Puerto Morelos, Mexico was conducted during August 2020 while active SCTLD was underway. SCTLD was first observed in Puerto Morelos in July 2018 (*Alvarez-Filip et al., 2019*). Surveillance was conducted from the sixth to the eighth night after the full moon and from 180 to 270 min after sunset based on spawning predictions for this species in the same region (http://132.248.121.70/Coral%20Spawning%20Predictions%20Puerto%20Morelos%202020.pdf). On the 7th night after the full moon, two of the 13 monitored colonies spawned: one colony with apparently 100% healthy tissue (Fig. 1A) and the other colony with more than 50% of its surface affected by SCTLD (Fig. 1B). At the time of gamete collection, the colony had multifocal areas of acute tissue loss. These areas of tissue loss were characterized by a central zone covered by an algal mat, sediment, and cyanobacteria, a transitional band covered by turf algae, and an outer band of bright white skeleton that bordered healthy tissue. Although no histological or molecular diagnostics were conducted, the gross signs both at the colony level and the community level (*i.e.,* the relative susceptibility of different species within the coral community) at the collection site were consistent with the case description of SCTLD. Also, the collection occurred within a time frame and geographical region where SCTLD was in a post-outbreak condition, with more than 10% prevalence, particularly for *P. strigosa*, a highly susceptible species to SCTLD (*Alvarez-Filip et al., 2022*).

Gamete bundles were collected with specially designed conical nets and taken to the laboratory within 30 min for fertilization and embryo culture (*Banaszak et al., 2018*) under permit PPF/DGOPA 070/20. A portion of the gamete bundles collected from this healthy colony were kept in contact in the same vial for at least 30 min after bundles broke up. Subsequent examination of these eggs (>60 min) revealed no self-fertilization and 100% sperm motility (See File S2) in this segregated vial of gametes, whereas full fertilization potential in this species is realized after only 15 min of gamete co-incubation (*Bennet et al., in review*).

An equal volume of gametes from each of the apparently healthy parent and diseased parent were mixed for one hour to facilitate fertilization. Fertilization was quantified after two hours using a dissecting microscope at 60X magnification (Motic SMZ-161, Kowloon, Hong Kong).

Embryos were rinsed three times with UVC-sterilized seawater previously filtered to 1 μm, and then divided among four indoor, static 'incubator' tanks (78 L each) with filtered and sterilized seawater, yielding an estimated 45,500 embryos per tank. The cultures were kept under static conditions for 72 h, with a partial water change (60%) after 48 h (*Banaszak et al., 2018*). Water temperature was controlled between 27° and 28 °C and salinity from 35 to 36 ppt, considered as the optimal parameters for fertilization, early development, and settlement in corals (*Vermeij, Fogarty & Miller, 2006*; *Randall & Szmant, 2009*).

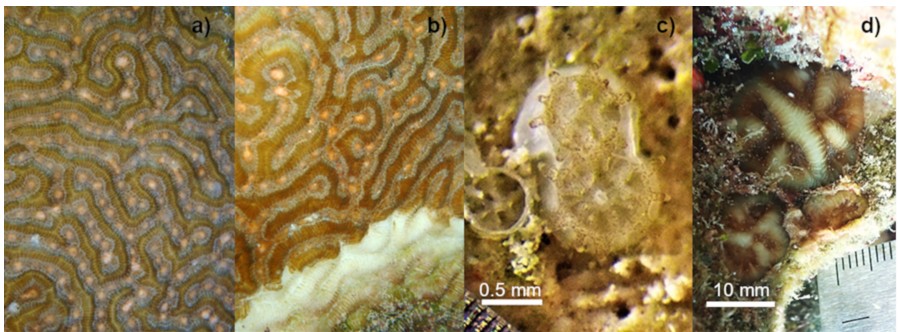

**Figure 1  Setting of gamete bundles in *Pseudodiploria strigosa* and recruits produced in 2020.** Setting of gamete bundles in *Pseudodiploria strigosa* prior to spawning of an apparently healthy colony (A) and a colony affected by SCTLD (B). Two-week-old *P. strigosa* recruits with symbionts and a skeleton (C). Sixteen-months-old *P. strigosa* recruits outplanted on the reef (D).

Once swimming larvae developed (~two days after fertilization), 100 concrete seeding units (SU: tetrapods-type II), designed by SECORE International (*Chamberland et al., 2017*) and previously conditioned in the sea for two months, were placed inside each incubator. Aeration was also added to oxygenate the cultures and generate slight water movement. After two weeks, larval settlement was recorded on a subset of substrates ($n = 20 - 30$ per tank), using a blue light lamp and yellow filter (Nightsea FL-1 light, Hatfield, PA, USA). The total number of settlers per tank was then extrapolated to estimate the settlement yield (% of original embryos converted to settlers) in each tank (*Miller et al., 2021*). The settlers were fed every two days with commercial supplements Koralle-VM and Coral Amino (Brightwell Aquatics, Fort Payne, AL, USA), as recommended by the manufacturer.

After one month, the seeding units with settlers were transferred to outdoor recirculating aquaria under controlled conditions of temperature (27–28 °C), salinity (35–37 ppt), and shaded natural light (350 and 480 $\mu$mol m$^{-2}$ s$^{-1}$). Seawater was partially replaced (30%) every third day. Recruits were fed twice a week with freshly hatched *Artemia* nauplii and once a week with commercial supplements Koralle-VM, Coral Amino and Liquid Reef (Brightwell Aquatics, USA, dosed as recommended by the manufacturer) for six months, after which commercial supplements were gradually reduced and feeding with *Artemia* nauplii was continued. The number of surviving recruits was recorded on a haphazardly selected subset of the settled substrates (from a larger population and different sample each time) at 4 ($n = 80$) and 13 ($n = 145$) months of age. The maximum diameter of recruits was derived from scaled photographs of a subset of nine tagged substrates during months 7, 9, and 10 ($n = 174$ recruits).

It should be noted that during this time, care was interrupted due to the impact of three hurricanes (Delta and Zeta in October 2020 and Grace in August 2021) on the Quintana Roo coast, which could have affected recruit growth and survival. After the passage of each cyclone, filamentous algae and ciliate blooms were observed due to sub-optimal conditions in the aquaria for a week after each event, leading to the death of some recruits during this

period (mortality was not quantified). In each case, the corals were treated promptly with commercial disinfectants (Microdyn, Tavistock Holdings, Baar Switzerland) with favorable results. Massive accumulations of *Sargassum* also affect local water quality (including the intake to the laboratory tanks), such as oxygen and pH reduction and eutrophication (*Van Tussenbroek et al., 2017*).

At 13 months of age, 145 seeding units with *P. strigosa* settlers were tagged and outplanted under permit SGPA/DGVS/04402/21 onto Jardines Reef, where apparently healthy adult *P. strigosa* colonies, as well as colonies with SCTLD were present. Three 10 m × 2 m transects were defined and seeding units were outplanted at a density of 4 substrates m$^{-2}$, avoiding sandy areas, unstable substrate, deep crevices, and aggressive invertebrates such as sponges, *Millepora* spp., *Erythropodium caribaeorum* and *Trididemnum solidum*. Each seeding unit was fixed with a nail and epoxy (Klipton Acuaplast, Mexico). Recruits were surveyed 0.5 and 3 months after outplanting. The size of live recruits on a subset of 42 substrates was also measured *in situ* (with a ruler) during these surveys. The substrates selected to obtain size measurements were those located along each transect in the quadrats aligned to meter 1, 6, and 10.

Similar gamete collection, fertilization, and settlement protocols of *P. strigosa* were undertaken in 2018, prior to SCTLD emergence in the region, based on spawn collected from three apparently healthy colonies at La Bocana Reef in Puerto Morelos. These parents were different to those used in 2020 because they died in 2019 during the SCTLD outbreak. Fertilization and larval settlement were performed similarly to 2020; this time, 50 preconditioned seeding units were placed in each of three incubator tanks. After two weeks, larval settlement was recorded on a subset of substrates ($n = 10$ per tank). The recruits were transferred to an outdoor recirculating aquaria system 20 days after settlement and outplanted to Jardines Reef shortly thereafter at one month of age. Seeding units were outplanted at the same density used in 2020, this time carefully wedging the seeding units into the reef crevices without adhesives or extra attachment materials. Given the difference in age of outplanting, results on post-outplant performance are provided only for rough qualitative comparison with those obtained in 2020. However, we compared larval settlement yield between 2018 and 2020 *via* a non-parametric Mann–Whitney U test (given the non-normality and high variability between replicates; *via* RStudio (*RStudio Team, 2020*)).

## RESULTS

The cross between a healthy and diseased colony produced healthy offspring, with high fertilization (95%) and settlement yield (22%) when compared to a batch cross among three healthy colonies obtained in a previous year (75% and 5%, respectively) and from other Caribbean regions (Table 1). Fertilization counts were not replicated. There was no significant difference in settlement yield between the 2020 cohort from one healthy and one diseased parent and the 2018 healthy-parent cross (Mann–Whitney U-test, $W = 0$, $p = 0.05183$). For larvae obtained in 2020, embryonic development and settlement appeared normal, similar to the process observed in 2018. There was a lack of malformed larvae and

**Table 1 Comparative table of settlement yields in *Pseudodiploria strigosa*.** Settlement yields of *Pseudodiploria strigosa* recruits produced from a cross between an apparently healthy colony and a colony affected by SCTLD that spawned in 2020 in Mexico (MEX) compared to those produced between healthy colonies that spawned in 2018 in Mexico prior to the SCTLD outbreak. Data are also compared to crosses among apparently healthy colonies from two other Caribbean regions, though culture conditions differed somewhat (described in *Miller et al. (2021)*: Curaçao (CUR) and Bahamas (BAH)). Scoring of settlement in the current study (2018 and 2020) was conducted ~2 weeks after exposure of substrates. Settlement yield was not significantly different between the two years (Mann-Whitney U-test, $p = 0.05$).

| Year | Location | Culture type | # Parent colonies | Est. # embryos /tank | % Fertilization | #SU /tank | # settlers/SU (mean ± SE) | Settlement yield (mean ± SE) |
|------|----------|--------------|-------------------|----------------------|-----------------|-----------|---------------------------|------------------------------|
| 2020[*] | MEX | Incubator tanks | 2 | 49,500 | 95 | 100 | 102 ± 8 | 22 ± 2.1 |
| 2018[*] | MEX | Incubator tanks | 3 | 45,600 | 70 | 50 | 48 ± 10.5 | 4.9 ± 4.1 |
| 2019[**] | CUR | CRIB | 6 | 700,000 | >90 | 720 | 87 ± 5.5 | 8.9 |
| 2018[**] | BAH | Aquaculture tanks | 2 | 60,000 | >90 | 200 | 43 ± 15 | 14.2 |

**Notes.**
[*]This study
[**](*Miller et al., 2021*)

CRIB, Coral Rearing In-situ Basins; SU, Seeding Unit; settlement yield, % of embryos converted to settlers (*Miller et al., 2021*).

the fully developed planula larva stage was observed 32 to 34 h after fertilization. Symbionts were observed around the mouth and in the tentacles of the polyps two weeks after settlement (Fig. 1C). After placement in outdoor aquaria at one month post-settlement, the survivorship of the 2020 cohort was 14.6% and 7.8%, at 4 and 13 months of age, respectively (Fig. 2A).

After seeding on the reef in 2020, the survival of recruits (relative to the number outplanted) was 60% after 15 days. The mean diameter of live recruits produced in 2020 increased over time from 0.8 mm ± 0.11 SD to 3.73 mm ± 1.94 SD during rearing in outdoor aquaria over 12 months (Fig. 2B). The mean diameter of these recruits increased to 6.5 mm ± 3.8 SD after three months on the reef (Figs. 1C–1D and 2B).

After three months, the 2018 recruits outplanted at 1-month old, had lower survival (10%) compared to the 2020 recruits sown at 13 months of age (44%) (Fig. 2A), likely due to their increased size/age at outplant rather than their parentage.

## DISCUSSION

This is the first record of the production of offspring from a colony of *Pseudodiploria strigosa* affected by SCTLD crossed with an apparently healthy colony. We show that the offspring can successfully complete settlement and initial growth *ex situ* as well as to establish themselves after outplanting onto a reef. Although it involved only a single diseased parental colony (crosses of two diseased colonies were not attempted), results disprove the null hypothesis that SCTLD prevents coral colonies such as *P. strigosa* from releasing mature and viable gametes. The high fertilization rate and settlement yield (proportion of initial embryos converted to outplantable settlers) of offspring from an affected colony compare favorably with other reported settlement yields on the same type of substrates in *ex situ* and *in situ* tanks (Table 1, *Miller et al., 2021*). It is higher compared to the settlement of larvae batch-crossed from three healthy colonies that occurred in the same lab in 2018 (4.8%) though variation in performance among cohorts of larvae is common (*Miller et al., 2018*; *Miller et al., 2021*). Similarly, we observed no adverse effects (malformation, unusual

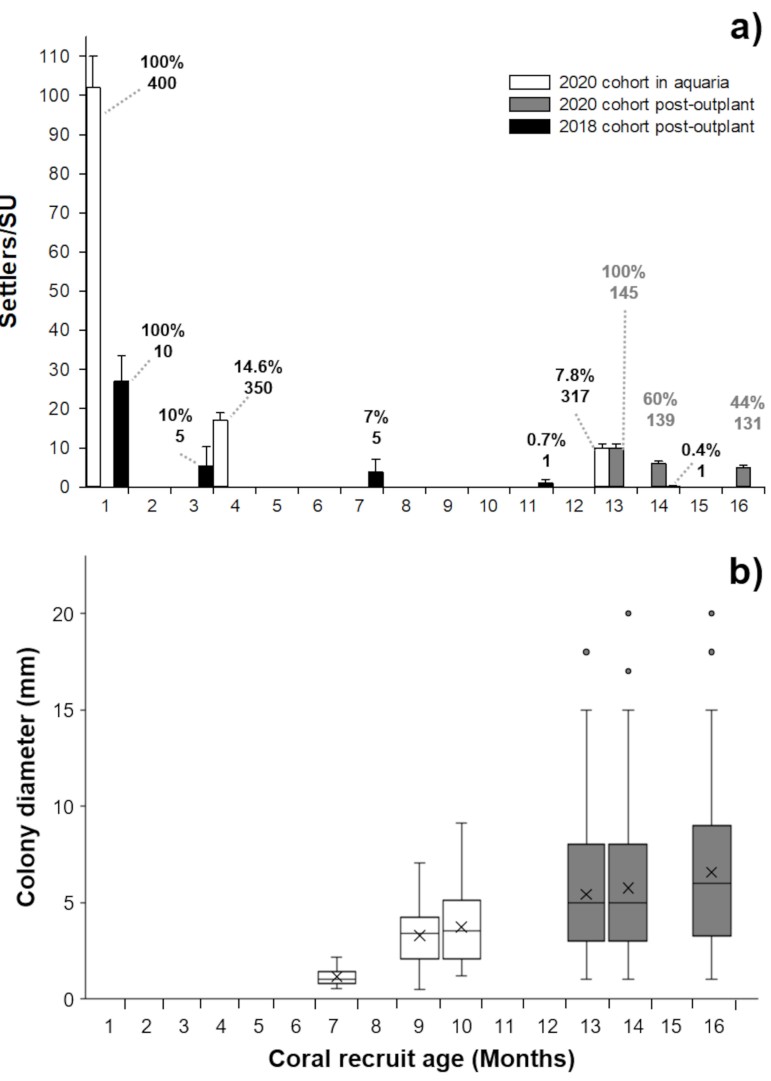

**Figure 2  Survivorship and size of *Pseudodiploria strigosa*.** (A) Mean (+1 SE) number of settlers per seeding unit (SU, $n = 400$) for the 2020 cohort produced from one healthy colony crossed with one colony affected by SCTLD. These settlers were kept in outdoor aquaria (solid white) and a subset were subsequently outplanted onto the reef at 13 months old (solid gray bars, $n = 145$) alongside a 2018 cohort that had been produced from apparently healthy parental colonies (solid black bars, $n = 10$) and outplanted to the reef at one month of age. Data shown in bars also include seeded units with zero survivors. Cumulative percent survival of recruits and the number of substrates retaining live recruits at each time point is given above each bar. (B) Maximum diameter of *P. strigosa* colonies from the 2020 cohort in outdoor aquaria (solid white) and after outplanting onto the reef (solid gray bars, $n = 45$). Error bars depict the 95% confidence interval, the bottom and top of the box are the 25th and 75th percentiles, the line inside the box is the 50th percentile (median), the cross is the average, and outliers are shown as open circles.

behavior or undue mortality) of diseased parentage on normal larval development and settlement.

Consistent with prior work showing a lack of vertical transmission of bacteria from parents *via* gametes and larvae in a range of Caribbean broadcast spawning corals (*Sharp*

*et al., 2010*), we saw no evidence that SCTLD pathogen(s) were transmitted vertically to offspring in *P. strigosa*. In contrast, *Sharp, Distel & Paul (2012)* did show vertical transmission of the microbiome to offspring in the brooding species, *Porites astreoides*, so further studies of vertical disease transmission of SCTLD in brooding corals may be warranted. On the other hand, *Williamson et al. (2022)* has recently shown that sexual recruits of two other brain corals are clearly susceptible to horizontal transmission of SCTLD. Hence appropriate biosecurity and care need to continue in terms of handling and housing SCTLD-affected colonies.

High rates of post-settlement mortality are expected for scleractinian corals (*e.g.*, *Arnold & Steneck, 2011*; *Trapon et al., 2013*; *Miller, 2014*). The post-outplant mortality we observed in both cohorts likely points to the period of acclimatization to the new natural environment, the absence of husbandry effort (*e.g.*, removal of filamentous algae, feeding) and control of environmental factors (*e.g.*, temperature, light, predation). The mortality of 2020 recruits we observed after outplanting on the reef was mainly due to fish predation, evidenced by bite marks affecting most recruits. This suggests that recruits, despite an extended husbandry effort for grow-out (13 months), were still susceptible to incidental predation by grazers (*Penin et al., 2010*). All substrates had accumulations of turf algae bound with sediment and benthic organisms began to colonize the seeding units as part of the succession process (*Arnold & Steneck, 2011*). Also, intact dead skeletons were observed, which may indicate mortality from disease (*Williamson et al., 2022*).

We document the increasing size distribution of 2020 recruits over time, both in tank culture and later on the reef. However, our data cannot distinguish growth of individual recruits from selective mortality of smaller-sized individuals. Although the variation in growth is wide, this observed pattern of size increase with reduction in mortality after these first few months on the reef (month 14–16, Fig. 2A) is a good indicator that these recruits are better able to compete with benthic organisms, resist predation, and reject sediment (*Babcock & Mundy, 1996*; *Vermeij & Sandin, 2008*) and can establish themselves within the natural reef site.

## CONCLUSIONS

We infer that SCTLD is not transmitted vertically through gametes since using gametes from a diseased colony did not affect fertilization, larvae development, settlement, or post-settlement success. It seems inevitable that the reduction in abundance and density of colonies from SCTLD will further impair natural recruitment of affected species. However, lack of vertical disease transmission to viable offspring demonstrates that assisted sexual reproduction can be applied in restoration efforts, even when it includes gametes from SCTLD-affected colonies.

## ACKNOWLEDGEMENTS

We thank Maria del Carmen García Rivas for her valuable support in the field. We also thank Fernando Negrete Soto, Miguel Ángel Gómez Reali, Edgar Escalante Mancera, José

Antonio Quintero Pérez, Eduardo Ávila Pech and Tania Doblado Speck for their technical assistance in the field and Edén Magaña Gallegos for aquarium support.

### Funding
This study was funded by the California Academy of Sciences to SECORE International, as well as the MAR Fund project numbers MX12-034 and MX13-033, CONACYT project number 425888, the Benito Juárez, Quintana Roo municipal government and project number 608 from the Marine Sciences and Limnology Institute of the National Autonomous University of Mexico (UNAM) to Anastazia T. Banaszak. The funders had no role in study design, data collection and analysis, decision to publish, or preparation of the manuscript.

### Grant Disclosures
The following grant information was disclosed by the authors:
California Academy of Sciences to SECORE International: MX12-034, MX13-033.
CONACYT: 425888.
Benito Juárez, Quintana Roo municipal government.
Marine Sciences and Limnology Institute of the National Autonomous University of Mexico (UNAM): 425888.

### Competing Interests
Margaret W. Miller is employed by SECORE International. Sandra Mendoza-Quiroz and Raúl Tecalco-Rentería are employed by SECORE International as contractors and Gandhi Germán Ramírez-Tapia was employed by SECORE International as a contractor. Anastazia Teresa Banaszak is an Academic Editor for PeerJ.

### Author Contributions
- Sandra Mendoza Quiroz conceived and designed the experiments, performed the experiments, prepared figures and/or tables, authored or reviewed drafts of the article, and approved the final draft.
- Raúl Tecalco Renteria performed the experiments, analyzed the data, authored or reviewed drafts of the article, and approved the final draft.
- Gandhi Germán Ramírez Tapia performed the experiments, analyzed the data, authored or reviewed drafts of the article, and approved the final draft.
- Margaret W. Miller analyzed the data, prepared figures and/or tables, authored or reviewed drafts of the article, and approved the final draft.
- Maria Victoria Grosso-Becerra performed the experiments, analyzed data, and approved the final draft.
- Anastazia T. Banaszak conceived and designed the experiments, authored or reviewed drafts of the article, and approved the final draft.

## Data Availability

The raw data is available in the Supplementary File.

## Supplemental Information

Supplemental information for this article can be found online at http://dx.doi.org/10.7717/peerj.15519#supplemental-information.

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
