# Peer review of "Coral affected by stony coral tissue loss disease can produce viable offspring"

_PeerJ, doi:10.7717/peerj.15519_

## Round 0.1 · original submission · Major Revisions

I am sorry for the delay in getting this decision back to you. I requested 4 reviews, and three came back with divergent opinions while the fourth responded that the manuscript was essentially unchanged from the initial version that was previously rejected. They felt that the criticisms were substantial and the fact that the authors did not incorporate their feedback from the initial submission merited rejection, but they did not provide any additional details and therefore I cannot evaluate their concerns here. Of the three referees that responded, two are quite positive and although both make numerous suggestions for improvements, they are relatively minor revisions to the manuscript for clarity or broader context. The final referee is more critical and has concerns about exactly how the authors confirmed the diagnosis of SCTLD, the statistical analyses and the availability of data as per the journal policy.

Overall, my impression is that these 3 referees are all supportive of a suitably revised manuscript, provided the authors can address the issues raised regarding disease diagnosis, statistical analyses, and data availability. In my view such concerns are substantial that may require additional review, so I am returning a decision of major revisions.

If you decide to incorporate the referee feedback as suggested, I ask that you provide a detailed cover letter outlining your response to each of the referee’s comments. I also point out that it is a common mistake made by many authors to respond to referee feedback in the letter only and not incorporate that same information into the manuscript itself. Future readers cannot see the response to the referees, and likely have similar questions, so we ask that your responses be reflected in the manuscript as well as the rebuttal letter. I look forward to seeing your responses and your revised manuscript.

·

Basic reporting

In general, I found the manuscript to be clear and well-reported. In the general comments section, I have provided a few suggestions for clarity as well as some references to be included/considered.

Experimental design

Though the sample size is quite low (just one cross), and the controls are from a different year and/or region, I do not believe this to be disqualifying. This represents the first known follow-through of reproduction of SCTLD individuals, and the authors do a good job of noting the caveats and of providing the context for the controls. The methods are well described, the question is addressed well, and they did a good job of following and tracking the individuals over a substantial length of time.

Validity of the findings

I find the results to be well-reported (with a few suggestions outlined in the general comments), and the discussion/conclusions to be substantiated.

Additional comments

I want to commend the authors on conducting this work and reporting it. Though it represents only a single example of an SCTLD-affected coral, it does confirm anecdotal reports of SCTLD corals spawning and, more importantly, follows that through the development, settlement, and juvenile stages to show that SCTLD-affected corals are capable of successful reproduction. This is a valuable contribution and, I believe, sets the stage for ongoing work with SCTLD-affected individuals both in and ex situ.
I have no major issues with the manuscript, but hope the notes below may help clarify some sections.
Lines 24-25: It wasn’t until the end of the introduction that I realized you had actually crossed an SCTLD-affected and apparently healthy coral together for this work. I would recommend rewording the abstract to clarify that there was a single cross between one parent of each AND also mention that you then compare the results of that to other studies/previous years and show it to be just as successful. You also don’t mention the size/growth information in the abstract, which you spend a fair bit of time on in your paper so may be worth mention. You may want to consider removing the Background/Methods/Results/Discussion headers from your abstract and just creating a paragraph that flows better and better represents the key points of your paper.

Line 38 and elsewhere (including references): The correct citation for what you currently have as FDEP 2018 is:
Florida Coral Disease Response Research & Epidemiology Team. Case Definition: Stony Coral Tissue Loss Disease. https://floridadep.gov/sites/default/files/Copy%20of%20StonyCoralTissueLossDisease_CaseDefinition%20final%2010022018.pdf; 2018

Lines 43-45: I find this sentence really unclear. For starters, coral density is not a %, so it can’t be reduced to less than 29% or 3%. I’m also not sure whether you are talking about reductions across all corals, or trying to highlight specific species. I am confused about the use of the FDEP 2019 website, particularly when so many peer-reviewed publications are available for these numbers.
Please reword paragraph for clarity. If you are trying to reference specific species, would recommend the literature (among others):
Precht, W.F.; Gintert, B.E.; Robbart, M.L.; Fura, R.; van Woesik, R. Unprecedented Disease-Related Coral Mortality in Southeastern Florida. Scientific Reports. 6:31374; 2016
Neely, K.L.; Lewis, C.L.; Lunz, K.S.; Kabay, L. Rapid Population Decline of the Pillar Coral Dendrogyra cylindrus Along the Florida Reef Tract. Frontiers in Marine Science. 8; 2021
Thome, P.E.; Rivera-Ortega, J.; Rodríguez-Villalobos, J.C.; Cerqueda-García, D.; Guzmán-Urieta, E.O.; García-Maldonado, J.Q.; Carabantes, N.; Jordán-Dahlgren, E. Local dynamics of a white syndrome outbreak and changes in the microbial community associated with colonies of the scleractinian brain coral Pseudodiploria strigosa. PeerJ. 9:e10695; 2021
Walton, C.J.; Hayes, N.K.; Gilliam, D.S. Impacts of a Regional, Multi-Year, Multi-Species Coral Disease Outbreak in Southeast Florida. Frontiers in Marine Science. 5; 2018

If you are instead trying to reference overall coral cover/density (not species-specific), I would recommend phrasing in terms of loss, not in terms of remaining coral cover, since that has high geographic variability even before SCTLD. References to consider for that might be:

Estrada-Saldívar, N.; Quiroga-García, B.A.; Pérez-Cervantes, E.; Rivera-Garibay, O.O.; Alvarez-Filip, L. Effects of the stony coral tissue loss disease outbreak on coral communities and the benthic composition of Cozumel reefs. Frontiers in Marine Science. 8:632777; 2021
Estrada-Saldıvar, N.; Molina-Hernández, A.; Pérez-Cervantes, E.; Medellın-Maldonado, F.; González-Barrios, F.J.; Alvarez-Filip, L. Reef-scale impacts of the stony coral tissue loss disease outbreak.
Walton, C.J.; Hayes, N.K.; Gilliam, D.S. Impacts of a Regional, Multi-Year, Multi-Species Coral Disease Outbreak in Southeast Florida. Frontiers in Marine Science. 5; 2018
Dahlgren, C.; Pizarro, V.; Sherman, K.; Greene, W.; Oliver, J. Spatial and Temporal Patterns of Stony Coral Tissue Loss Disease Outbreaks in The Bahamas. Frontiers in Marine Science:767; 2021
Heres, M.M.; Farmer, B.H.; Elmer, F.; Hertler, H. Ecological consequences of stony coral tissue loss disease in the Turks and Caicos Islands. Coral Reefs. 40:609-624; 2021

Lines 58 and 60: Can you be specific about which diseases these colonies were experiencing in these studies?

Line 70: Recommend adding the following reference which describes the rescue efforts (rather than just the spawning efforts)
Neely, K.L.; Lewis, C.L.; O'Neil, K.; Woodley, C.M.; Moore, J.; Ransom, Z.; Moura, A.; Nedimyer, K.; Vaughan, D. Saving the last unicorns: the genetic rescue of Florida's pillar corals. Frontiers in Marine Science. 8:876; 2021

Lines 73-74: Recommend adding the following references as other lab transmission experiments
Aeby, G.; Ushijima, B.; Campbell, J.E.; Jones, S.; Williams, G.; Meyer, J.L.; Hase, C.; Paul, V. Pathogenesis of a tissue loss disease affecting multiple species of corals along the Florida Reef Tract. Frontiers in Marine Science. 6; 2019
Studivan, M.S.; Baptist, M.; Molina, V.; Riley, S.; First, M.; Soderberg, N.; Rubin, E.; Rossin, A.; Holstein, D.M.; Enochs, I.C. Transmission of stony coral tissue loss disease (SCTLD) in simulated ballast water confirms the potential for ship-born spread. 2022

Line 84: The word “currently” will not age well in a publication. Consider rewriting to note when SCTLD appeared at this reef and when the study was done so it is clear SCTLD was present without providing a time stamp that may not match when the reader is accessing the article.

Line 146-147: Please provide some quantitative data here. How high was fertilization? How high was settlement yield? What were some values from previous years. I know you show this in table 1, but mention of it in the results is also important.

Line 149: comma should be before and, not before symbionts

Lines 156-157 are presumptive, stating why they were more resistant. Do you have data to support this? Or citations? Consider either making this a hypothesis, or backing it up somehow.

Line 161: There is no figure 2c

Line 164-165: I am unclear what “affected by SCTLD being incorporated by settlement and initial growth ex situ” means. Please reword for clarity.

Line 192-193: This information on the P. asteroides study needs more context. What sort of bacterial transmission occurred in that study? Disease? Microbiome? All bacteria? This seems like an interesting comparison, but it is sort of thrown into the first sentence of the conclusions with no context. Maybe instead put into discussion? And definitely provide some more information on what this is, why it is relevant.

Lines 273-276: All of your references are standardized by being non-capitalized in their titles except this one. Consider revising for clarity.

Line 289: elkhorn coral should not be capitalized.

Table 1 caption: Last line seems to be missing a closing parenthesis.

Table 1: Please ensure the asterisk information (starting with *This study) is all below the table rather than smushed into the side.

Reviewer 2 ·

Basic reporting

This paper is mostly clearly written and has conformed to the PeerJ standards, and the raw data are supplied. The background information successfully introduces most of the concepts required for the reader to understand the work, however I believe it could be strengthened by adding an introduction to the study species. I suggest adding some detail on the reproductive ecology and importance to the reef community of this species, as well as reasoning for its use in this study. There was one paragraph in the Results section (see Line Edits in the "General Comments" section, for detail) that I found quite confusing, but that I believe can be made clearer with revision. The literature cited is relevant and appropriate. The figures are high quality, however I recommend some revisions to Figure 2a, in particular. As currently presented, Figure 2a seems to focus on the number of settlers per settlement unit over time, across different spawning events. My understanding of the current work is that settlers per settlement unit is not a key part of the analysis and I recommend this figure be rearranged to focus on settler survival, to align with the primary findings reported in the results. Further, I recommend that the discussion be expanded to discuss existing points more thoroughly and include some other topics that should be introduced, such as a clearer statement of where the field stands (or stood, prior to this report) regarding transmission of stony coral tissue loss disease (SCTLD) and/or similar diseases between parents and offspring. In my opinion, this would also give the authors an opportunity to more clearly define the step the current research takes and how it advances the field.

Experimental design

This paper presents original work that falls within the Scope of the journal, and it addresses a clearly defined research question that fills an important knowledge gap in the field. The investigation is well-designed with thorough monitoring, and the methods are described with sufficient detail to replicate. The only exception to this concerns the 2018 spawning event, which I believe should be described in greater detail to help the reader understand the similarities and differences between the 2018 and 2020 larval rearing conditions (details of this suggestion can be found in the Line Edits included with this review in the "General Comments" section). I do recommend some revisions, including addressing whether any self-fertilization controls were conducted. I appreciate that self-fertilization is rare in scleractinians that bundle their gametes, but since your conclusions rest on the fact of the larvae being the products of both parents, some data/references to that effect would be useful. Additionally, I’m concerned by the assumption of SCTLD vs other diseases with similar presentations by gross observation alone. The need to address this assumption by either confirming SCTLD or acknowledging that alternatives cannot be ruled out is a key improvement for this manuscript going forward.

Validity of the findings

While I believe these results to be of high value and I am strongly supportive of this work, I have some doubts about the validity of the findings as currently presented. I am, however, optimistic that my concerns can be put to rest with some meaningful additions. My greatest issue with the findings is that they rest on the assumption that one parent of the larvae studied was healthy while the other was affected by SCTLD, but it does not appear that this disease diagnosis was confirmed histologically or otherwise. As mentioned in the previous review section, without a confirmation of the diagnosis beyond gross observation, the findings remain questionable. Additionally, in order to accept these findings I believe statistical analysis is required, especially to support statements comparing quantitative metrics that describe different groups of larvae. Supporting the findings with statistical work, even if statistically significant relationships are not identified, would be a substantive improvement that would make the findings more robust. Finally, I believe the data from the 2018 event should also be included in your raw data files, as I gather that this manuscript will be the first publication of those data, and especially as they form a keystone axis of comparison to the results obtained from the 2020 spawn. Generally, as the major comparison in this paper seems to be between the 2018 and 2020 spawns (healthy x healthy and healthy x diseased), more information about the 2018 spawn is needed. As is, the 2018 recruits are mentioned, but few data or methods are presented except in comparison to the 2020 recruits. I think the comparison is appropriate and useful, but this paper would be strengthened by adding more information about the 2018 event to support it.

Additional comments

Line edits for “Coral affected by stony coral tissue loss disease can produce viable offspring”

L20: I recommend changing the word “dispersion.” If I understand correctly, you’re referring to the distance/spatial separation between colonies. Although dispersion may technically be correct, it could lead to confusion with “dispersal” i.e. a larval activity that at this point in your paper the reader doesn’t yet know you aren’t discussing given the subject matter at hand. Maybe just “separation” would work?
L22-23: this sentence could be revised to speak more plainly, which would make it easier for the reader to understand.
L31: “rescue” is strong – maybe replace or add “restoration” to provide more measured scope
L46: Could you add a bit more to link the process of calcification to the decline in coral density?
L49: “are known to” rather than “often” – this would better set up the following sentences that detail relevant examples.
Paragraph starting L49: good and clear statement of current knowledge
L76: Suggest this be rephrased to be more specific that the lack of evidence you refer to is related to post-embryonic survival. Alternatively add a bit more to explain why the information you present in the previous paragraph regarding O. faveolata and A. palmata does not address “reproductive viab[ility]” in the way you use the term here.
L79: it seems the only thing missing from comparison (healthy/healthy in 2018, healthy/SCTLD in 2020) is SCTLD/SCTLD – I think that would be useful to mention in the Discussion section at least, if not here as well.
L89-90: can you be sure that colonies were actually healthy/diseased?
L126: I don’t think “several” is sufficient here, could you add an estimate? I understand this wasn’t quantified in the study, but it would be good to know if these mortality events included substantial numbers relative to recruits in the aquaria at that time. Also, maybe “some” is better than “several,” as, to me at least, “several” means 3-7 (unless that’s the actual estimate you make, in which case adding that and keeping “several” would address my concern).
L129: Could you describe the impact to water quality of the Sargassum accumulations? Higher nutrient levels? Turbidity? How was the water quality perturbed?
L149: Could you elaborate here or in the Methods on how you determined that development progressed normally – perhaps what information you used as a point of reference? Or simply the fact that most/all of the embryos successfully developed into larvae?
L151: I think “at” is more appropriate than “after” here
L152: I think you can delete “calculated” and “at” on this line.
L153: Again, I don’t think you need “calculated” and this could just read “…the survival relative to total recruits outplanted decreased…”
Paragraph starting L151: As written, I find this paragraph pretty difficult to follow in terms of the comparison between different instances of outplanting and relative percentages of survival/mortality. I think it can be improved but would need a heavy revision. So there were two outplantings – one in 2020 (the result of the cross described in this paper) and one in 2018 – that differed in their rearing conditions – from the 2020 clutch in which recruits were reared in outdoor aquaria from 1mo post settlement and outplanted to the reef at 13mo, while the 2018 clutch was outplanted at 1mo (?). These were monitored for survival at 15 days post seeding and three months post seeding? Also, I’m quite confused by the switch from percent survival to percent mortality and which numbers refer to which groups/what the percentages are out of. I believe I understand the first sentence, essentially describing survival of recruits that were settled for one month and then moved to outdoor aquaria before they experienced different experimental treatments. In the sentence starting “After three months,” are the mortality rates again relative to the number of outplanted recruits? Would it be possible to stick with percent survival throughout rather than switching to mortality? Also, I would find it useful to see a survival rate, maybe in parentheses, showing their survival at these stages relative to the original number of larvae so that the reader could get a sense of how the numbers have whittled down over time.
L157: I recommend this final phrase, presenting reasons that the older recruits had better survival, be moved to the discussion and also suggest its tone be adjusted. Even if you include references to works that found these things to be true in other cases, unless you have data to support the variables you list, this is speculation – I happen to agree with your thinking, but still, it should not be stated as fact. I wonder if the word “likely” might be a useful addition to address my comment, i.e. “…since due to their size, age and symbiosis, they were likely more resistant to reef environmental changes and predation.”
L168-169: These data regarding fertilization rate and settlement yield belong in the Results section, as does the information about settlement rate from 2018.
L173: Could you add a bit more about what adverse effects you didn’t see? Maybe what milestones/metrics you were looking for to confirm that the diseased parentage hadn’t affected the larvae?
L175: Please add references to support this statement regarding post-settlement mortality.
L175-6: Similar to a previous comment, unless you have data to truly assert this, I suggest measuring the tone a bit more to make it clear that this is your hypothesis to explain the post-outplant mortality. If you have references you can add to support your thinking, those would also be welcome here.
Paragraph starting L175: This paragraph moves quickly through several topics that I think could be expanded, including references. Unless space is an issue, I highly recommend exploring some of these to a greater degree. For example, as a reader I wonder if there are other comparable species or studies that might add more context to the growth rates you report – “somewhat lower than” the one study is a start, but what about other records, especially disparities between growth rates of aquarium-held and wild populations. Even if none exist, that too would be relevant to report. Another opportunity would be to discuss the dynamics of mortality as corals move through their life stages, for example providing and building on a reference for your comment (L182-3) that “such mortality is typical of early stage recruits…”
L192: I think you need only use “infer” or “apparently” – using both reduces the strength of the statement.
L194: Does any research exist about reproduction from two diseased colonies? That strikes me as a useful data point to make this statement, or at least to acknowledge as a next step.
L194-195: I’d really like to see some statistical analysis in the results to support this “does not affect” assertion.

FIGURE 2:
A) I appreciate that this figure has to incorporate a lot of disparate data, but I’m confused by the arrangement, which highlights settlers/SU as the main variable for analysis – from the results section, it seems that survivability is the major takeaway from this paper. Currently, this plot mostly appears to compare the utilization of SUs over time and between groups, and survivability is harder to tease out. I find the x-axis very helpful to keep the different groups straight over time/age, but I think the y-axis should be changed, maybe to survival rate, with information in the caption to explain how it was calculated (i.e. relative to recruits placed in outdoor aquaria, until outplanting when survival rate was recalculated to refer to the number of recruits originally outplanted). I think the percent or number of SUs could be indicated much as the percent survival is currently, just as a number associated with each bar.
B) I wonder if a set of trendlines might be useful to really distinguish the trajectory of the in-aquaria growth rate from that of the outplants.

TABLE 1
Missing closing parenthesis at end of caption. The definitions of the asterisk/two asterisk indications should be in the caption, not the table itself.
This table is a good start, but I believe it could be improved. Some thoughts:
1. Were all the settlement yield measurements/counts conducted at the same time relative to time on the reef and/or age of recruits? If not, this would seem to have great confounding potential and should be addressed. Please clarify in the caption.
2. Would like to see some analysis of whether the numbers from the 2021 diseased/healthy cross were significantly different from those of the healthy/healthy crosses in 2018 by this group. An indication of statistical relationships among the data would strengthen your arguments.
3. In addition to the clarification from point (1), a clearer statement of when each measurement was made, e.g. at what point was the number of settlers/SU recorded, at least for the work by this group in 2018 and 2021, either in the results text and/or indicated in the table caption would help the reader interpret the comparison among groups.

·

Basic reporting

With a few minor exceptions that I addressed in the line-by-line suggestions, I found this manuscript to be very well-written, clear, well-structured, and sufficiently supported by referenced literature. The primary exception to this is that I feel the authors should add some background information and context about the general reproductive strategy, etc. of the study species. I found the submission self-contained regarding addressing hypotheses presented.

Experimental design

I found the research question well defined, and extremely relevant at this time given that I believe it is the first study to begin to examine the impacts of SCTLD on coral reproduction, and as the authors mentioned in general the impact of disease on coral reproduction is extremely minimal in general. Methods were sufficient in my opinion to recreate this study if desired.

Validity of the findings

I think the authors did an excellent job overall at keeping claims and hypotheses tempered in light of the small sample size, however I think they should soften their conclusion regarding SCTLD transmission, as detailed in my line-by-line comments. All relevant data seem to be provided.

Additional comments

Abstract
Line 20: Suggest changing to “increased *relative* dispersion, or “decreased density” before “…of the colonies reduces fertilization potential..”. Do this elsewhere as needed too.
Line 24: Think it should be clearer that the authors are referring to gametes from each of these colonies that were then cross-fertilized, I’m not sure what the proper terminology is but when first reading I wasn’t sure what the reproductive strategy of P. strigosa was, so this was not immediately apparent to me.
Introduction
General: I recommend adding a brief section giving an overview of the reproductive strategies of Pseudodiploria strigosa, i.e. I believe they are sequential hermaphrodites with male and female colonies, spawn once yearly, and are broadcast spawners (all from Szmant 1986, 10.1007/BF00302170) just to help give readers some context before they start reading your experimental methods. Also mention whether they inherit symbionts from the vertical or horizontal transmission.
Additionally, this may be a personal preference, but I find references included mid-sentence tend to interrupt the flow of the reader. Unless it is a complex statement like the one on lines 49-52, I recommend putting references at the end of the statement rather than right after the portion of the sentence they are supporting.
Line 39: Just as an FYI, the Florida Department of Environmental Protection along with other regulating bodies in Florida have shifted to calling it “Florida’s Coral Reef” instead of the Florida Reef Tract. Sources below.
https://floridadep.gov/rcp/rcp/content/floridas-coral-reefs
https://www.nature.org/en-us/about-us/where-we-work/united-states/florida/stories-in-florida/floridas-spectacular-coral-reef-system/
https://floridascoralreef.org/

Line 41: The sources the authors have currently do not support the almost 30 corals affected by SCTLD statement, the highest count is in Alvarez-Filip et al. 2019 which lists 24. I’ve heard higher numbers mentioned and have no doubt that in the wider Caribbean this is true, but it needs to be backed up by a source. Recommend just sticking with the 22 number the authors have in the abstract
Line 49: Recommend a transition sentence here, i.e. “In addition to impacting the cover and diversity of coral colonies on the reef, diseases may impair the recovery potential of those corals remaining as well by impacting reproductive success” or similar.
Line 74: Typo in reference, should be “Meiling” not “Melling”
Line 78: Add comma after “Pseudodiploria strigosa”
Materials & Methods
Line 99: Assuming these are similar to conditions from where the gametes were collected from? Recommend stating if so.
Line 111: I recommend not using an acronym for “seeding units” since it’s only used a few times. Also, SU is used to reference “seeding units” here but in Table 1 “SU” is used to reference “substrate units” I know they are probably effectively the same thing (I think) but make sure to be consistent.
Results
Line 156: I am unsure what you mean in this sentence by “symbiosis”, please clarify.
Discussion
General: Without getting too much into speculation, I wonder if the authors have any idea why they might have seen higher fertilization and settlement rates from the SCTLD x healthy parent gametes when compared to their 2020 healthy gametes. Could be worth addressing in the discussion briefly.
Lines 181-183: Can you add a citation to back up this statement? The part “mortality is often typical of early stage recruits”.
Conclusions
Line 192: There are several factors in this experiment that may have affected offspring to parent transmission of SCTLD, including the removal of larvae from the SCTLD-endemic reef environment, rinsing them, and moving them to UVC-sterilized and filtered water. Perhaps this diluted whatever agent is responsible for SCTLD enough to prevent later disease becoming apparent. Or perhaps many of the settled larvae that died were actually affected by SCTLD. I think an actively SCTLD-affected colony producing viable gametes is a hugely significant enough finding in itself, and the focus should primarily be on that.

---

## Round 0.2 · accepted · Accept

Thank you for your detailed revisions and responses to the referees - you have addressed everyone's concerns with the initial submission, and all three referees are supportive of the acceptance of this revised manuscript. There are a few minor suggestions in terms of tense or the like that you may want to incorporate, but I feel this version is close enough to move into production.

·

Basic reporting

I had no issues with the basic reporting in the first manuscript, and this manuscript is even improved upon that.

Experimental design

I had no issues with the experimental design in the first manuscript, and this manuscript is even improved upon that. The sample size is very small, but not disqualifying, and is important as a first step in these novel observations.

Validity of the findings

I had no issues with the validity of the findings in the first manuscript, and this manuscript is even improved upon that.

Additional comments

I would like to again commend the authors on this work and on the revisions which have improved it.
I have only a few very minor comments and suggestions, mostly for clarity.

Line 39 (“SCTLD has spread”): I would suggest this will not continue to age with the manuscript. Perhaps rephrase to “By 2023, SCTLD had spread…”

Line 58: Recommend removing comma between marginal and lesions

Lines 60-61: It is not customary to capitalize disease names (yellow band and white plague)

Line 63: This states that egg volume was lower in colonies with partial mortality. Is this standardized to tissue area? I would think that a colony that had lost 50% of its tissue would inherently have lost 50% of its egg volume compared to its pre-death state. Is this reduction referring to this? Or was the egg volume within the remaining tissue also reduced?

Line 97 (“with more than 50% of its surface affected by SCTLD”): Please clarify. Does this mean 50% recent mortality? The other explanation is that 50% of the surface area was an active lesion, but that seems unlikely.

Line 101-106: I particularly like this description of how SCTLD status was determined. Though histology has been suggested by some to be a “defining feature” of SCTLD, it has not really been compared with other diseases to show conclusive differences. I am happy to see that you have defined this in a way that makes sense in the field.

Line 104: You may want to consider referencing the case definition here:
Florida Coral Disease Response Research & Epidemiology Team. Case Definition: Stony Coral Tissue Loss Disease. https://floridadep.gov/sites/default/files/Copy%20of%20StonyCoralTissueLossDisease_CaseDefinition%20final%2010022018.pdf; 2018

Line 111: I appreciate the description of the self-fertilization control here.

Line 235: should the word “that” be between organisms and began? Sentence unclear.

Reviewer 2 ·

Basic reporting

This paper does a nice job of reporting on an important issue with a sufficient depth of references and a clear articulation of the issue at hand, how this study addresses the issue, and what conclusions can be drawn. The raw data is shared appropriately, and the figures and tables are well put together.

Experimental design

Very well designed study that used a combination of available and experimental data to address an important, timely issue in the field. The Methods are described sufficiently and the authors take care to carefully address any confounding variables, which strengthens the results.

Validity of the findings

The findings of this study are highly valid, and, as stated above, extremely timely and useful to other researchers in the field.

Additional comments

I have a few line edits to add, but these are extremely minor. Well done!

L58: remove comma after (marginal)
L123-24: recommend adding “shallow-water” or “tropical” or similar before “corals,” as these conditions are not necessarily ideal for all corals (e.g. mesophotic or cold-water).
L198: Remove comma after “old”

·

Basic reporting

The minor issues I had with the author's conclusions have been sufficiently addressed.

Experimental design

No comment.

Validity of the findings

The minor issues I had with the author's conclusions have been sufficiently addressed.

Additional comments

I commend the authors on their thorough job of addressing mine and other reviewer's comments. Looking forward to seeing this published.